# SayTap: Language to Quadrupedal Locomotion

**Yujin Tang**
yujintang@google.com
Google DeepMind

**Wenhao Yu**
magicmelon@google.com
Google DeepMind

**Jie Tan**
jietan@google.com
Google DeepMind

**Heiga Zen**
heigazen@google.com
Google DeepMind

**Aleksandra Faust**
sandrafaust@google.com
Google DeepMind

**Tatsuya Harada**
harada@mi.t.u-tokyo.ac.jp
The University of Tokyo

**Abstract:** Large language models (LLMs) have demonstrated the potential to perform high-level planning. Yet, it remains a challenge for LLMs to comprehend low-level commands, such as joint angle targets or motor torques. This paper proposes an approach to use foot contact patterns as an interface that bridges human commands in natural language and a locomotion controller that outputs these low-level commands. This results in an interactive system for quadrupedal robots that allows the users to craft diverse locomotion behaviors flexibly. We contribute an LLM prompt design, a reward function, and a method to expose the controller to the feasible distribution of contact patterns. The results are a controller capable of achieving diverse locomotion patterns that can be transferred to real robot hardware. Compared with other design choices, the proposed approach enjoys more than 50% success rate in predicting the correct contact patterns and can solve 10 more tasks out of a total of 30 tasks. (https://saytap.github.io)

**Keywords:** Large language model (LLM), Quadrupedal robots, Locomotion

## 1 Introduction

Simple and effective interaction between human and quadrupedal robots paves the way towards creating intelligent and capable helper robots, forging a future where technology enhances our lives in ways beyond our imagination [1, 2, 3]. Key to such human-robot interaction system is enabling quadrupedal robots to respond to natural language instructions as language is one of the most important communication channels for human beings. Recent developments in Large Language Models (LLMs) have engendered a spectrum of applications that were once considered unachievable, including virtual assistance [4], code generation [5], translation [6], and logical reasoning [7], fueled by the proficiency of LLMs to ingest an enormous amount of historical data, to adapt in-context to novel tasks with few examples, and to understand and interact with user intentions through a natural language interface.

The burgeoning success of LLMs has also kindled interest within the robotics researcher community, with an aim to develop interactive and capable systems for physical robots [8, 9, 10, 11, 12, 13]. Researchers have demonstrated the potential of using LLMs to perform high-level planning [8, 9], and robot code writing [11, 13]. Nevertheless, unlike text generation where LLMs directly interpret the atomic elements—tokens—it often proves challenging for LLMs to comprehend low-level robotic commands such as joint angle targets or motor torques, especially for inherently unstable legged robots necessitating high-frequency control signals. Consequently, most existing work presume the provision of high-level APIs for LLMs to dictate robot behaviour, inherently limiting the system's expressive capabilities.

We address this limitation by using foot contact patterns as an interface that bridges human instructions in natural language and low-level commands. The result is an interactive system for legged robots, particularly quadrupedal robots, that allows users to craft diverse locomotion behaviours flexibly. Central to the proposed approach is the observation that patterns of feet establishing and breaking contacts with the ground often govern the final locomotion behavior for legged robots due

7th Conference on Robot Learning (CoRL 2023), Atlanta, USA.

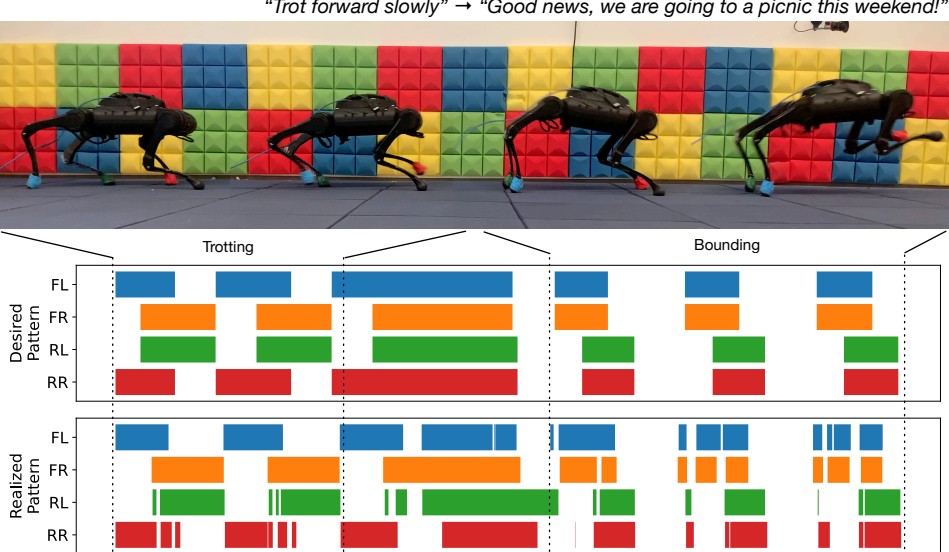

Figure 1: Illustration of the results on a physical quadrupedal robot. We show two test commands at the top, and the snapshots of the robot in the top row of the figure. The row in the middle shows the desired contact patterns translated from the commands by an LLM (the pattern in between the commands requests the robot to put all feet on the ground and stand still), and the bottom row gives the realized patterns. The proposed approach allows the robot to take both simple and direct instructions (e.g., "Trot forward slowly") as well as vague human commands (e.g., "Good news, we are going to a picnic this weekend!") in natural language and react accordingly.

to the heavy reliance of quadruped locomotion on environmental contact. Thus, a contact pattern, describing the contact establishing and breaking timings for each legs, is a compact and flexible interface to author locomotion behaviors for legged robots. To leverage this new interface for controlling quadruped robots, we first develop an LLM-based approach to generate contact patterns, represented by '0's and '1's, from user instructions. Despite that LLMs are trained with mostly natural language dataset, we find that with proper prompting and in-context learning, they can produce contact patterns to represent diverse quadruped motions. We then develop a Deep Reinforcement Learning (DRL) based approach to generate robot actions given a desired contact pattern. We demonstrate that by designing a reward structure that only concerns about contact timing and exposing the policy to the right distribution of contact patterns, we can obtain a controller capable of achieving diverse locomotion patterns that can be transferred to the real robot hardware.

We evaluate the proposed approach on a physical quadruped robot, Unitree A1 [14], where it successfully controls the robot to follow diverse and challenging instructions from users (Figure 1). We benchmark the proposed approach against two baselines: (1) using discrete gaits, and (2) using sinusoidal functions as interface. Evaluations on 30 tasks demonstrate that the proposed approach can achieve 50% higher success rate in predicting the correct contact pattern and can solve 10 more tasks than the baselines.

The key contributions of this paper are: i) A novel interface of contact pattern for harnessing knowledge from LLMs to flexibly and interactive control quadruped robots; ii) A pipeline to teach LLMs to generate complex contact patterns from user instructions; iii) A DRL-based method to train a low-level controller that realizes diverse contact patterns on real quadruped robots. Finally, our proposal also holds intriguing potential for both human-robot interaction researchers and the robotic locomotion community, inviting a compelling cross-disciplinary dialogue and collaboration.

## 2 Related Work

### 2.1 Language to robot control

There is a rich literature in leveraging language to modulate the behavior of robots [15, 10, 8, 16, 17, 18, 19, 20]. Earlier work in this direction typically assumes structured text templates to translate language to robot commands [17, 19] or leveraged natural language processing (NLP) tools such as

the parse tree to assist extracting the constraints from user input, followed by trajectory optimization to obtain robot motion [20]. Though these approaches demonstrate complex robotics tasks, they usually do not handle unstructured natural language input. To mitigate this issue, recent work leverages the advancements in representation learning and deep learning to train language conditioned policies that mapped unstructured natural language instructions to robot actions [18, 21, 22, 23]. To establish proper mappings between natural language embeddings and robot actions, these approaches usually require a significant amount of demonstration data with language labels for training the policy, which is challenging to collect for diverse legged locomotion behaviors.

Inspired by recent success in LLMs to perform diverse tasks [5, 6, 7], researchers in robotics have also explored ideas to connect LLMs to robot commands [8, 9, 11, 12, 13, 24, 25]. For example, Ahn et al. [8] combined LLMs with a learned robot affordance function to pick the optimal pre-trained robot skills for completing long horizon tasks. To mitigate the requirement for pre-training individual low-level skills, researchers also proposed to expand the low-level primitive skills to the full expressiveness of code by tasking LLMs to write robot codes [11, 12, 13]. As LLMs cannot directly generate low-level robot motor commands such as joint targets, these approaches had to design an intermediate interface for connecting LLMs and robot commands, such as high-level plans [8, 9, 24], primitive skills [11, 12], and trajectories [25]. In this work, we identify foot contact patterns to be a natural and flexible intermediate interface for quadrupedal robot locomotion that do not require laborious design efforts.

## 2.2 Locomotion controller for legged robots

Training legged robots to exhibit complex contact patterns, especially gait patterns, has been extensively studied by researchers in robotics, control, and machine learning. A common method is to model the robot dynamics and perform receding horizon trajectory optimization, i.e., Model Predictive Control (MPC), to follow desired contact patterns [26, 27, 28, 29, 30]. For quadruped robots, this led to a large variety of canonical locomotion gaits such as trotting [26], pacing [31], bounding [32], and galloping [33], as well as non conventional gaits specified by the desired contact timing or patterns [28, 30]. Despite the impressive results in these work, applying MPC to generate diverse locomotion behavior often requires careful design of reference motion for robot base and swing legs and high computational cost due to re-planning. Prior work have also explored using learning-based methods to author flexible locomotion gaits [34, 35, 36, 37, 38, 39, 40]. Some of these work combines learning and MPC-based methods to identify the optimal gait parameters for tasks [34, 35, 36]. Others directly train DRL policies for different locomotion gaits, either through careful reward function design [37, 40], open-loop commands extracted from prior knowledge about gaits [38, 39] or encoding of a predefined family of locomotion strategies that solve training tasks in different ways[41]. This paper explores an alternative DRL-based method that relies on the simple but flexible reward structure. Compared to the prior work, the proposed reward structure only concerns about contact timing thus is more flexible in generating diverse locomotion behaviors.

## 3 Method

The core ideas of our approach include introducing desired foot contact patterns as a new interface between human commands in natural language and the locomotion controller. The locomotion controller is required to not only complete the main task (e.g., following specified velocities), but also to place the robot's feet on the ground at the right time, such that the realized foot contact patterns are as close as possible to the desired ones, Figure 2 gives an overview of the proposed system. To achieve this, the locomotion controller takes a desired foot contact pattern at each time step as its input, in addition to the robot's proprioceptive sensory data and task related inputs (e.g., user specified velocity commands). At training, a random generator creates these desired foot contact patterns, while at test time a LLM translates them from human commands.

In this paper, a desired foot contact pattern is defined by a cyclic sliding window of size $L_w$ that extracts the four feet ground contact flags between $t + 1$ and $t + L_w$ from a pattern template and is of shape $4 \times L_w$. A contact pattern template is a $4 \times T$ matrix of '0's and '1's, with '0's representing feet in the air and '1's for feet on the ground. From top to bottom, each row in the matrix gives the foot contact patterns of the front left (FL), front right (FR), rear left (RL) and rear right (RR) feet. We demonstrate that the LLM is capable of mapping human commands into foot contact pattern templates in specified formats accurately given properly designed prompts, even in cases when the

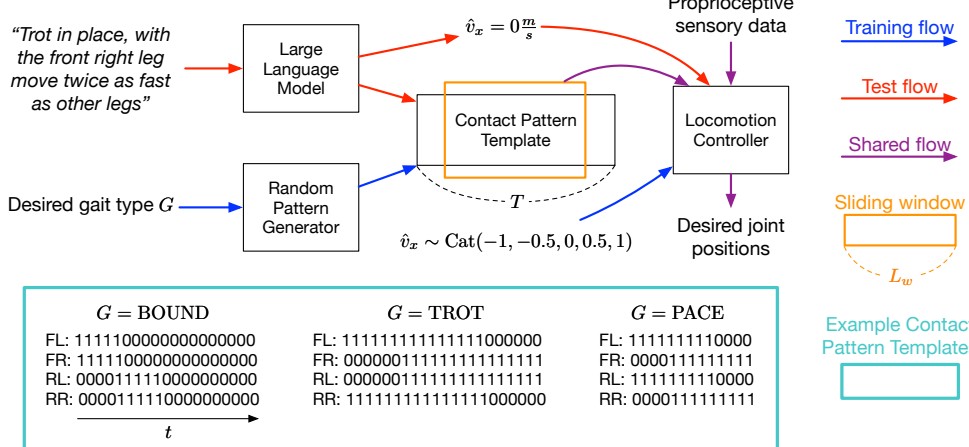

Figure 2: Overview of the proposed approach. In addition to the robot's proprioceptive sensory data and task commands (e.g., following a desired linear velocity $\hat{v}_x$), the locomotion controller accepts desired foot contact patterns as input, and outputs desired joint positions. The foot contact patterns are extracted by a cyclic sliding window of size $L_w$ from a pattern template, which is generated by a random pattern generator during training, and is translated from human commands in natural language by an LLM in tests. We show some examples of contact pattern templates at the bottom.

commands are unstructured and vague (Section 3.1). In training, we use a random pattern generator to produce contact pattern templates that are of various pattern lengths $T$, foot-ground contact ratios within a cycle based on a given gait type $G$ (Section 3.2.2), so that the locomotion controller gets to learn on a wide distribution of movements and generalizes better.

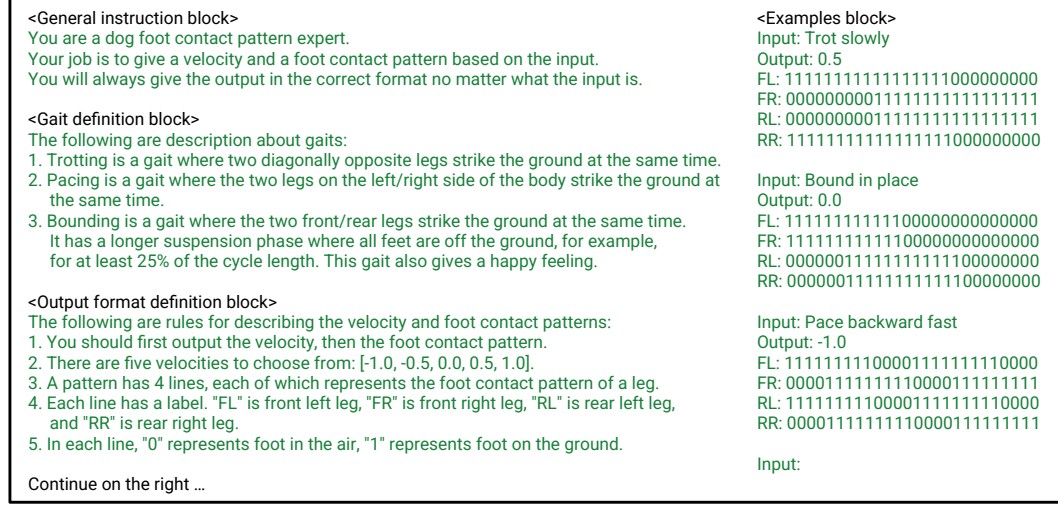

Figure 3: Our exact prompt for our method in all experiments. The final "Input:" is followed by user specified command. Texts in black are for explanation and are not used as input to the LLM.

## 3.1 Language to Foot Contact Patterns

Although LLMs can learn knowledge from a vast amount of text data at training, providing proper prompts at inference is the key to unlock and direct the acquired knowledge in meaningful ways. Carefully designed prompts serve as the starting point for the models to generate text and guide the direction and context of the outputs. The proposed approach aims to enable the LLM to map any human commands in natural language to foot contact patterns in a specified format. Figure 3 lists the prompts used in this paper, wherein we group them into four categories:

1. **General instruction** describes the task the LLM should accomplish. In this paper, the LLM is expected to translate an arbitrary command to a foot contact pattern. Note that examples of such translations will be provided in **Examples block**.

2. **Gait definition** gives basic knowledge of quadrupedal gaits. Although their descriptions are neither exhaustive nor sufficiently accurate, experimental results suggest that it provides enough information for the LLM to follow the rules. It also connects the bounding gait to a general impression of emotion. This helps the LLM generalize over vague human commands that do not explicitly specify what gaits the robot should use.

3. **Output format definition** specifies the format of the output. We discretize the desired velocities $\hat{v}_x \in \{-1, -0.5, 0, 0.5, 1\}\frac{m}{s}$ so that the LLM can give proper outputs corresponding to commands that contain words like "fast(er)" and "slow(er)".

4. **Examples block** follows the general knowledge of instruction fine-tuning and shows the LLM a few concrete input-output pairs. Although we give the LLM three commonly seen gait examples only, experimental results show that it is able to generalize and handle various commands, including those vaguely state what velocity or gait the robot should use.

## 3.2 Foot Contact Pattern to Low-level Commands

### 3.2.1 Problem Formation

We formulate locomotion control as a Markov Decision Process (MDP) and solve it using DRL algorithms. An MDP is a tuple $(S, A, r, f, P_0, \gamma)$, where $S$ is the state space, $A$ is the action space, $r(s_t, a_t, s_{t+1})$ is the reward function, $f(s_t, a_t)$ is the system transition function, $P_0$ is the distribution of initial states $s_0$, and $\gamma \in [0, 1]$ is the reward discount factor. The goal of a DRL algorithm is to optimize a policy $\pi : S \mapsto A$ so that the expected accumulated reward $J = E_{s_0 \sim P_0}[\sum_t \gamma^t r(s_t, a_t, s_{t+1})]$ is maximized. Here, $a_t = \pi_\theta(s_t)$ and $\theta$ is the set of learnable parameters. In locomotion tasks, $s_t$ often includes sensory data and goal conditions (e.g., user specified velocity commands [42]), and $a_t$ is desired joint angles or motor torques. We expand $s_t$ to include a desired foot contact pattern, and the controller needs to achieve the main task as well as realize the desired foot contact patterns.

### 3.2.2 Random Pattern Generator

The random pattern generator receives a gait type $G$, it then randomly samples a corresponding cycle length $T$ and the ground contact ratio within the cycle for each feet, conducts proper scaling and phase shifts, and finally outputs a pattern template. Due to the space restrictions, we defer the detailed implementation and illustrations in the Appendix. While humans can give commands that map to a much wider set of foot contact pattern templates, we define and train on five types: $G \in \{\text{BOUND, TROT, PACE, STAND\_STILL, STAND\_3LEGS}\}$. Examples of the first three types are illustrated at the bottom of Figure 2, the latter two types are trivial and omitted in the figure.

### 3.2.3 Locomotion Controller

We use a feed-forward neural network as the control policy $\pi_\theta$. It outputs the desired positions for each motor joint and its input includes the base's angular velocities, the gravity vector $\vec{g} = [0, 0, -1]$ in the base's frame, user specified velocity, current joint positions and velocities, policy output from the last time step, and desired foot contact patterns. In this paper, we use Unitree A1 [14] as the quadrupedal robot. A1 has 3 joints per leg (i.e., hip, thigh and calf joints) and $L_w = 5$ in all experiments, therefore the dimensions of the policy's input and output are 65 and 12, respectively. The policy has three hidden layers of sizes $[512, 256, 128]$ with $\text{ELU}(\alpha = 1.0)$ at each hidden layer as the non-linear activation function.

To encourage natural and symmetric behaviors, we employ a double-pass trick in the control policy which has been shown to be effective in other scenarios too [43, 44]. Specifically, instead of using $a_t = \pi_\theta(s_t)$ directly as the output, we use $a_t = 0.5[\pi_\theta(s_t) + f_{\text{act}}(\pi_\theta(f_{\text{obs}}(s_t)))]$, where $f_{\text{act}}(\cdot)$ and $f_{\text{obs}}(\cdot)$ flips left-right the policy's output and the robot's state respectively. Intuitively, this double-pass trick says the control policy should output consistently when it receives the original and the left-right mirrored states. In practice, we find this trick greatly improves the naturalness of the robot's movement and helped shrink the sim-to-real gap.

### 3.2.4 Task and Training Setups

The controller's main task is to follow user specified linear velocities along the robot's heading direction, while keeping the linear velocity along the lateral direction and the yaw angular velocity as close to zeros as possible. At the same time, it also needs to plan for the correct timing for feet-ground

strikes so that the realized foot contact patterns match the desired ones. For real world deployment, we add a regularization term that penalizes action changing rate so that the real robot's movement is smoother. In addition to applying domain randomization, we find that extra reward terms that keep the robot base stable can greatly shrink the sim-to-real gap and produce natural looking gaits. Finally, although no heavy engineering is required to train the locomotion policy with extra contact pattern inputs, we find it helps to balance the ratio of the gait types during training. Please refer to the Appendix for hyper-parameters and detailed settings.

## 4 Experiments

We conducted experiments to answer three questions. Throughout the experiments, we used GPT-4 [45] as the LLM. Please see the Appendix for experimental setups.

### 4.1 Is Foot Contact Pattern a Good Interface?

The first experiment compares foot contact pattern with other possible interface designs. One option is to introduce intermediate parameters as the interface, and have the LLM map from human natural language to the parameter values. We use two baseline approaches for comparison: Baseline 1 contains a discrete parameter $G$ that is the 5 gait types introduced in Section 3.2.2; Baseline 2 contains 4 tuples of continuous parameters $(a_i, b_i, c_i), i \in \{1, 2, 3, 4\}$ that defines a sinusoidal function $y_i(t) = \sin(a_i t + b_i)$ and its cutoff threshold that defines the foot-ground contact flag for the $i$-th foot – FOOT_ON_GROUND $= \mathbb{1}\{y_i(t) \leq c_i\}$. Here, $t \in [1, T]$ is the time step within the cycle. We construct foot contact pattern templates based on the values output by the LLM (e.g., for Baseline 1, we use the random pattern generator; for Baseline 2, we use the sinusoidal functions and the cutoff values) and check if they are correct.

Figure 4 shows the prompts for the two baselines, where they are based on the prompt in Figure 3 with necessary modifications. Table 1 gives the commands we use in this experiment; commands 1–20 are basic instructions that express explicitly what the robot should do, whereas commands 21–25 test if the interface design allows generalization and pattern composition. We set GPT-4's temperature to 0.5 to sample diverse responses, and for each approach we submit each command five times. For each submission, we use the top-1 result only for comparisons.

We implement domain knowledge based checker programs for each command for objective evaluations (see Appendix D), and we summarize the results in Figure 5. Aggregating over all the commands and test trials, the proposed approach gets significantly ($\sim 50\%$) higher accuracy than the baselines (see the left-most plot in the first row). Despite of having only three conventional examples in the context, the LLM almost always maps the human commands correctly to the expected foot contact patterns. The only exception in the test commands is command 21, where the LLM is correct only one out of the five tests. It mostly fails to generate columns of 0s in the pattern template, but

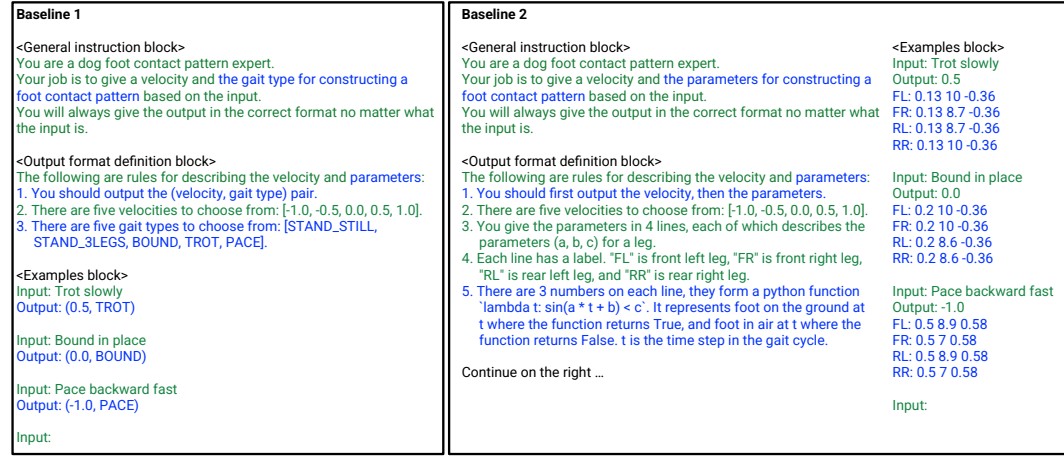

Figure 4: Baselines prompts. Differences from our prompt are highlighted in blue. The "Gait definition block" is not changed and omitted in the figure. Texts in black are for explanation thus they are not used as input to the LLM.

Table 1: Commands for generated pattern template evaluation. We observed the foot contact patterns generated by the LLM after accepting the commands, and compared them against our checkers.

| Id | Command |
|---|---|
| 1 | Stand still |
| 2–5 | Lift front left / front right / rear left / rear right leg |
| 6–8 | Bound / Trot / Pace in place |
| 9–11 | Bound / Trot / Pace forward slowly |
| 12–14 | Bound / Trot / Pace forward fast |
| 15–17 | Bound / Trot / Pace backward slowly |
| 18–20 | Bound / Trot / Pace backward fast |
| 21 | Trot in place, with a suspension phase where all feet are off the ground |
| 22 | Trot forward, with the front right leg moving at a higher frequency |
| 23 | Stand on front right and rear left legs |
| 24 | Walk with 3 legs, with the rear right foot always in the air |
| 25 | Bound then pace, you can extend the pattern length if necessary |

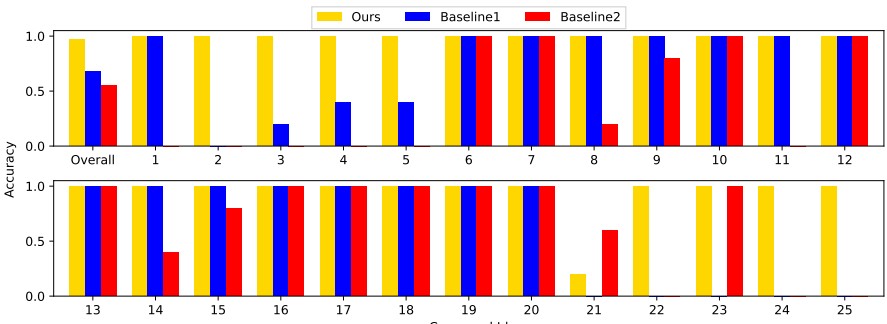

Figure 5: Accuracy comparison of generated patterns. For each command in Table 1, we generate 5 patterns from the LLM and compare them against the expected results. We show the aggregated accuracy over all commands on the left of the first row, followed by the individual results.

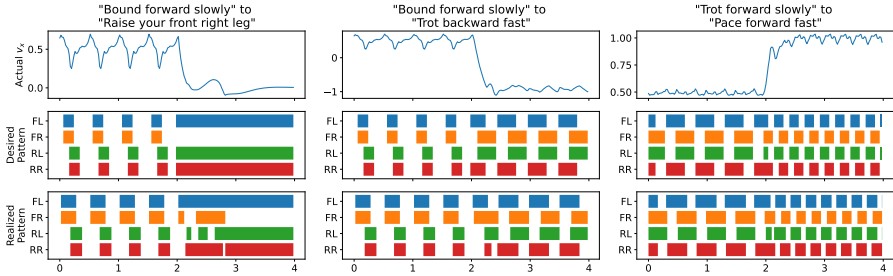

Figure 6: Velocity tracking and foot contact pattern realization in simulation. We show the actual linear velocity along the robot's heading direction (first row), the desired foot contact pattern (middle row) and the realized foot contact pattern (last row) from three test trials. The commands given to the robot in each trial are shown at the top of the plots.

in one interesting case, it appends an extra row of "S: 00⋯0" to the pattern template, trying to convince us of the existence of the required suspension phase. Baseline 1 gets the second highest accuracy; it achieves high scores on the basic instructions but fails completely for commands 21–25. Considering that this is how we sample patterns and train the controller, these results are somewhat expected. It fails to generate the correct patterns for commands 2–5 because the random pattern generator selects randomly a foot to lift for $G$ = STAND_3LEGS. Although we could have relaxed the design of Baseline 1 so that it accepted extra parameters for $G$, we didn't have to do so for the proposed approach and it still worked out. Moreover, this design modification has very limited effect and highlights the restrictions imposed by these high-level APIs. Unlike Baseline 1, Baseline 2 should have sufficient freedom in the parameter space to handle all the commands (maybe not command 25), yet its overall accuracy is the worst. Although we performed prompt engineering and constructed the examples carefully in its context for Baseline 2, the LLM has difficulty in understanding the relationship between gaits and the underlying mathematical reasoning. This limitation

again highlights the motivation and demonstrates the importance of the proposed approach. The experimental results indicate that foot contact pattern is a good interface as it is both straightforward and able to provide more flexibility in the human command space.

## 4.2 Can we learn to accomplish the main task and realize the contact pattern?

Following [42], we train the locomotion controller with the Proximal policy optimization (PPO) [46] in the IsaacGym simulator [47]. The controller's main task is to track a user specified linear velocity along the robot's heading direction $v_x$, and at the same time, to place the feet correctly to produce the desired foot contact patterns. Figure 6 shows the results in simulation. The commands given to the robot in each trial are shown at the top of the plots. It can be seen from the figure that the controller learns to track the specified velocity (e.g., "slow"/"fast" corresponds to $0.5\frac{m}{s}/1.0\frac{m}{s}$ in absolute values) and manages to place the robot's feet correctly to produce foot contact patterns that are close to the desired ones. Furthermore, we successfully transfer the learned controller and deploy it on the physical robot without any fine-tuning. Figure 1 gives some analytical results on the physical robot. Please watch the accompanying video for the motions.

## 4.3 Does the proposed approach work with unstructured/vague instructions?

The proposed approach enables both the quadrupedal robot to follow direct and precise commands and unstructured and vague instructions in natural language that facilitates human robot interactions. To demonstrate this, we sent commands in Table 2 to the robot and observe its reactions. Note that unlike in the previous tests, none of the human expressions here stated explicitly what the robot should have done or what gait it should have used. Based on the subjective evaluation, the observed motions were capable of expressing the desired emotion (e.g., jumping up and down when excited) and presenting the scene accurately (e.g, struggling to move when we told that it had a limping leg), the reactions were mostly consistent with the expectations. This will unlock many robot applications, ranging from scene acting and human companion to more creative tasks in industries and homes.

Table 2: Extended tests. The commands in this test do not tell the robot explicitly what it should do.

| Command | Observed Robot Motion |
|---|---|
| Good news, we are going to a picnic! | Jumping up and down |
| Back off, don't hurt that squirrel! | Moving backward slowly in trotting gaits |
| Act as if the ground is very hot | Pacing fast, with its feet barely touching the ground |
| Act as if you have a limping rear left leg | Struggling to walk, with its RL leg hardly moving |
| Go catch that squirrel on the tree | Bounding fast forward toward the prey |

## 5 Conclusions

This paper devised an interactive system for quadrupedal robots that allowed users to craft diverse locomotion behaviours flexibly. The core ideas of the proposed approach include introducing desired foot contact patterns as a new interface between natural language and the low-level controller. During training, these contact patterns are generated by a random generator, and a DRL based method is capable of accomplishing the main task and realizing the desired patterns at the same time. In tests, the contact patterns are translated from human commands in natural language. We show that having contact patterns as the interface is more straightforward and flexible than other design choices. Moreover, the robot is able to follow both direct instructions and commands that do not explicitly state how the robot should react in both simulation and on physical robots.

**Limitations and Future Work**

One limitation of the proposed approach is that domain knowledge and trial-and-error tests are necessary to design the random pattern generator, such that the patterns used for training are feasible. Furthermore, while increasing the variety of the random patterns would essentially increase the locomotion capability of the robot, training on a large set of gaits is hard since it involves the trade-off of sample balancing and data efficiency.

One may train a set of expert policies separately, where each of which specializes in one motion, then use imitation learning to distill the experts to address this problem. Another interesting direction for future works is to modify the current pattern representation and make it more versatile (e.g., replacing 0s and 1s with 0s and $H$'s to specify desired foot clearance $H$), alternatively methods in [48, 49] can also be incorporated to achieve the same effect.

**Acknowledgments**

The authors would like to thank Tingnan Zhang, Linda Luu, Kuang-Huei Lee, Vincent Vanhoucke and Douglas Eck for their valuable discussions and technical support in the experiments.

The experiments in this work were performed on GPU virtual machines provided by Google Cloud Platform.

This work was partially supported by JST Moonshot R&D Grant Number JPMJPS2011, CREST Grant Number JPMJCR2015 and Basic Research Grant (Super AI) of Institute for AI and Beyond of the University of Tokyo.

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

# A  More about Random Pattern Generator

Given a specified gait $G$, there are 4 steps for the random pattern generator to create a template, and Figure 7 illustrates the process when $G$ = PACING. To acquire knowledge as to what ranges of settings are feasible for the robot, we first train the robot in simulation for simple locomotion tasks (i.e., follow desired linear velocities). We then analyze the learned gait (the agents seem to learn exclusively trotting, probably because the reward design in those libraries) and measure the ranges. To generate a template in general, we

- [Step 1] Sample template length $T$. In our implementation, $T \in [24, 28]$, since the control frequency is 50 Hz, this corresponds to a cycle length of $0.48 \sim 0.56$ seconds.
- [Step 2] Sample a foot-ground contact length ratio within the cycle $r_{\text{contact}} \in [0.5, 0.7]$. $Tr_{\text{contact}}$ therefore gives the number of '1's and $T(1 - r_{\text{contact}})$ the number of '0's in each row.
- [Step 3] Scale cycle length and ground contact ratio. This only applies to $G \in \{\text{BOUND}, \text{PACE}\}$ because these two gaits require shorter foot contact to make it natural and dynamically more feasible. For $G$ = BOUND, we shorten the foot-ground contact time to 60% of the sampled value (i.e., $r_{\text{contact}} = 0.6r_{\text{contact}}$); For $G$ = PACE, we keep $r_{\text{contact}}$ untouched, but shrink the cycle length to half its sampled value (i.e., $T = 0.5T$).
- [Step 4] Shift patterns for corresponding legs. This step requires domain knowledge of quadrupedal locomotion and is gait type dependent. For example, for $G$ = BOUND, we place the ones at the beginning of the FL and FR rows and shift those in the RL and RR rows by $0.5Tr_{\text{contact}}$ bits to the right; For $G$ = PACE, we place the ones at the beginning in the FL and RL rows and at the end of the FR and RR rows.

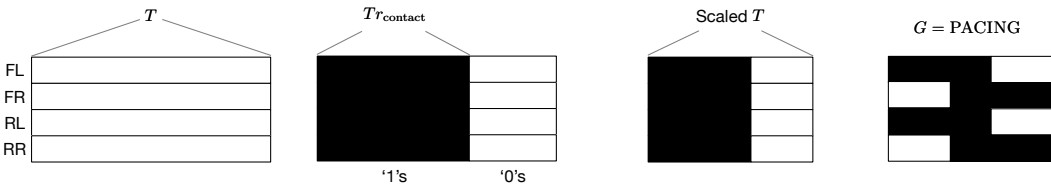

Figure 7: How the random pattern generator works.

# B  Reward Design

Our reward design is based on those in legged gym [42]. The total reward consists of 8 weighted reward terms: $J = \sum_{i=1}^{8} w_i r_i$, where $w_i$'s are the weights and $r_i$'s are the rewards. The definition of each reward term and the value of the weights are in the following. We put the purpose of each reward term in the bracket at the beginning of the description.

- [Task Reward] Linear velocity tracking reward. $r_1 = e^{-4 \times ((v_x - \hat{v}_x)^2 + v_y^2)}$, where $v_x$ and $\hat{v}_x$ are the current and desired linear velocities along the robot's heading direction, and $v_y$ is the current linear velocity along the lateral direction. All velocities are in the base frame, and $w_1 = 1$.
- [Task Reward] Angular velocity tracking reward. $r_2 = e^{-4 \times \omega_z^2}$, where $\omega_z$ is the current angular yaw velocity in the base frame and $w_2 = -0.5$.
- [Task Reward] Penalty on foot contact pattern violation. $r_3 = \frac{1}{4} \sum_{i=1}^{4} |c_i - \hat{c}_i|$, where $c_i, \hat{c}_i \in \{0, 1\}$ are the realized and desired foot-ground contact indicators for the $i$-th foot, and $w_3 = -1$.
- [Sim-to-Real] Regularization on action rate. $r_4 = \sum_{i=1}^{12} (a_t - a_{t-1})^2$ where $a_t$ and $a_{t-1}$ are the controller's output at the current and the previous time steps, and $w_4 = -0.005$.
- [Sim-to-Real] Penalty on roll and pitch angular velocities. We encourage the robot's base to be stable during motion and hence $r_5 = \omega_x^2 + \omega_y^2$, where $\omega_x$ and $\omega_y$ are the current roll

and pitch angular velocities in the base frame. This penalty does not apply to $G = $ BOUND and $w_5 = -0.05$.

- [Sim-to-Real] Penalty on linear velocity along the z-axis. Similar to the previous term, we use this term to encourage the base stability during motion. $r_6 = v_z^2$ where $v_z$ is the current linear velocity along the z-axis in the base frame. This penalty does not apply to $G = $ BOUND either and $w_6 = -2$.

- [Natural Motion] Penalty on body collision. $r_7 = \sum_{i=1}^{K} \mathbb{1}\{F_i > 0.1\}$, where $F_i$ is the contact force on the $i$-th body. In our experiments $K = 8$ (i.e., 4 thighs and 4 calves) and $w_7 = -1$.

- [Natural Motion] Penalty on deviation from the default pose. $r_8 = \sum_{a_t \in \text{hip}} |a_t|$, where $a_t$'s are the actions (i.e., deviation from the default joint position) applied to the hip joints, and $w_8 = -0.03$.

## C  Training Configurations

### C.1  Control

We use PD control to convert positions to torques in our system. The bases value for the 2 gains are $k_p = 20$ and $k_d = 0.5$. Our control frequency is 50 Hz.

### C.2  Gait Sampling

We randomly assign a gait $G$ to a robot at environment resets, and also samples it again every 150 steps in simulation. Of the 5 $G$'s, some gaits are harder to learn than others. To avoid the case where the hard-to-learn gaits die out, leaving the controller to learn only on the easier gaits, we restrict the sampling distribution such that the ratio of the 5 $G$'s are always approximately the same.

### C.3  Reinforcement Learning

We use the Proximal policy optimization (PPO) [46] algorithm as our reinforcement learning method to train the controller. In our experiments, PPO trains an actor-critic policy. The architecture of the actor is introduced in Section 3.2.3, and the critic has the identical network architecture except that (1) its output size is 1 instead of 12, and (2) it also receives the base velocities in the local frame as its input. We keep all the hyper-parameters the same as in [42] and train for 1000 iterations. For safety reasons, we end an episode early if the body height of the robot is lower than 0.25 meters. Training can be done on a singe NVIDIA V100 GPU in approximately 15 minutes.

### C.4  Domain Randomization

During training, we sample noises $\epsilon \sim$ Unif, and add them to the controller's observations. We use PD control to convert positions to torques in our system, and domain randomization is also applied to the 2 gains $k_p$ and $k_d$. Table 3 gives the components where noises $\epsilon$ were added and their corresponding ranges.

Table 3: Domain randomization settings.

| # | Component | Noise Range |
|---|---|---|
| 1 | Base linear velocities | $[-2, 2]$ |
| 2 | Base angular velocities | $[-0.25, 0.25]$ |
| 3 | Gravity vector in the base frame | $[-1, 1]$ |
| 4 | Joint positions | $[-1, 1]$ |
| 5 | Joint velocities | $[-0.05, 0.05]$ |
| 6 | $k_p$ | $[-5, 0]$ |
| 7 | $k_d$ | $[0, 0.25]$ |

# D Objective Evaluation on Generated Patterns

We implemented a domain knowledge based check program for each of the commands in Table 1, and evaluated the generated patterns with these checkers to produce Figure 5. By domain knowledge, we mean knowledge about quadrupedal locomotion as to what each gait pattern should look like (e.g., the robot should move its diagonal legs together when trotting, while in pacing gait the robot should move legs on the left/right side of the body together, etc).

