# OpenReview forum: "SayTap: Language to Quadrupedal Locomotion"
_robot-learning.org/CoRL/2023/Conference — CoRL 2023 Poster_

### Official Review · Reviewer_erUF · 2023-07-16

**Confidence:** 4
**Originality:** Good
**Technical Quality:** Very Good
**Clarity Of Presentation:** Excellent
**Impact:** 3

**Recommendation:**

Weak Accept: I recommend accepting the paper, but will not argue for my recommendation if the majority of other reviewers have a different opinion.

**Review:**

The paper is well-written and interesting to read. The topic of grounding large language models to control robotic devices is very important for the future of robotics. The paper clearly states the task and describes the developed method. The proposed interface for interaction between LLM and locomotion controller is original and works better than the baselines.

Strengths: Authors proposed a simple and quite effective interface that bridges the gap between language models and locomotion controllers. The developed method was evaluated on a real quadruped robot.

Weaknesses: The expressive power of foot contact patterns may be limited for complex motions like jumping or climbing stairs.


**Quality Of The Limitations Section:**

Limitations are addressed clearly

**Questions For Rebuttal:**

In the paper, LLM  was given only five gate patterns to choose from (BOUND, TROT, PACE, STAND_STILL, STAND_3LEGS). Would it be possible for LLM to generate a new meaningful gate pattern with a sufficient explanation in the prompt?


**Robotics Focus:**

Sufficient demonstration on hardware

**Summary Of Paper:**

The paper considers the task of controlling a quadrupedal robot with natural language commands. The authors propose to use foot contact patterns as an interface between a large language model and a locomotion controller. They contribute prompts for LLM, a reward function to train a locomotion controller, and an algorithm to generate feasible contact patterns. The authors benchmark the proposed interface against two baselines, in particular discrete gaits and sinusoidal functions. The proposed approach achieves more than a 50% success rate and can solve more tasks then the baselines.

**Summary Of Recommendation:**

I recommend accepting the paper. It considers an important task of grounding LLM to control a physical robot. The proposed method is novel and achieves good results.

---

### Official Review · Reviewer_Ztd6 · 2023-07-19

**Confidence:** 5
**Originality:** Very Good
**Technical Quality:** Good
**Clarity Of Presentation:** Good
**Impact:** 4

**Recommendation:**

Weak Accept: I recommend accepting the paper, but will not argue for my recommendation if the majority of other reviewers have a different opinion.

**Review:**

Strengths --
- High relevance: The problem of constructing an interface from natural language to motor skills is very relevant and timely. This paper's formulation is thought-provoking and will serve as a concrete artifact to advance the discussion.
- High originality: This is the first paper I've seen that meaningfully connects LLMs to the motor skills of a quadruped. As a researcher working with similar tools, I did not think of constructing the prompt/interface this way myself. After reading this paper, my understanding of the possibilities and limitations of motor skill-related prompting has expanded.
- Clarity: The paper is clearly written.

Weaknesses --
- Low generality: It is essential that the (contact schedule + velocity) grammar is both (a) very low-dimensional compared to the space of motor policies and (b) spans the space of natural language descriptions well. These criteria are met through careful engineering incorporating domain knowledge specific to quadruped locomotion on flat ground. I expect these criteria could be difficult to satisfy for other domains and tasks.
- Low direct utility: The system is mostly useful for walking in expressive locomotion styles. This is useful for entertainment but does not improve the ability of the robot to (a) move from place to place or (b) interact with its environment. If the primary downstream use case is entertainment, it would strengthen the paper to evaluate this more thoroughly. Currently, the only supporting evidence is the qualitative self-evaluation in Table 2.

**Quality Of The Limitations Section:**

Additional details required

**Questions For Rebuttal:**

Add experiments showing failure cases to mark the system limitations and facilitate follow-up work. The prompts in Table 2 are chosen intentionally to work well using the proposed method. However, the space of interesting natural language commands for quadrupeds is much larger. It would be nice to explicitly include some failure cases and describe what the robot does. I can think of lots of totally reasonable "unstructured/vague instructions" for a quadruped that would fail and explicitly illustrate limitations:

- "Roll over" -> outside of grammar
- "Climb on top of a box directly in front of you" -> outside of grammar
- "Balance on one foot" -> dynamically infeasible
- "Kick the ball" -> not enough sensory information
- "Bring me a Coke" -> not enough sensory information

It's possible to reason from first principles that these prompts won't succeed due to the above. But for a casual reader, including them would help make the limitations explicit. Also, reporting those failure cases will help justify future work that accomplishes them! And, I'm curious to know what the system will actually do in these cases (maybe it'll be different than what I expect?)

**Robotics Focus:**

Sufficient demonstration on hardware

**Summary Of Paper:**

Large language models are increasingly powerful for tasks expressed in natural language. However, they are not helpful for directly generating robot motor commands, which are thought to have a substantially different structure than natural language. This paper proposes an intermediate grammar for specifying quadruped robot motions. This grammar is the robot's foot contact pattern and body velocity. Through reinforcement learning in simulation, a mapping is established between this grammar and a motor-level control policy. Then, a large language model is prompted to translate natural language into this grammar. The combined pipeline can adjust the robot control policy in response to natural language. For unstructured or vague instructions such as "Good news, we are going to a picnic!" or "Go catch that squirrel on the tree" the robot's behavior is subjectively reasonable.

**Summary Of Recommendation:**

This paper will be a valuable landmark in the important discussion surrounding grounding natural language in motor skills. I encourage expanding the evaluation to include failure cases which will make the limitations concrete and guide follow-up works.

---

### Official Review · Reviewer_8oYu · 2023-07-20

**Confidence:** 4
**Originality:** Good
**Technical Quality:** Good
**Clarity Of Presentation:** Good
**Impact:** 3

**Recommendation:**

Weak Accept: I recommend accepting the paper, but will not argue for my recommendation if the majority of other reviewers have a different opinion.

**Review:**

## Strengths And Weaknesses

The chosen interface is interesting as it seems to be quite versatile as it should cover a large number of gaits.
The pipeline with the RL controller also looks like an interesting building block towards useful applications.

The idea is interesting and results on real robots are very welcome, but some points need to be clarified/improved.

The evaluation is not clear.
It says "We observed the foot contact patterns generated by the LLM after accepting the commands and compared them against our expectations".
Is it a subjective evaluation? is it a visual evaluation or do you have some kind of edit distance between the generated and expected foot patterns?
It would be interesting to show a failure case.
The evaluation also questions the need for an LLM.
If a human can easily generate and compare a foot contact pattern, why would an LLM be needed? (since its output might also be unreliable).
I agree that the RL controller is still useful in this case.

It seems that the cycle duration is almost constant (T in [24, 28]), could the authors comment on this?
How was the cycle duration range chosen?
Same question for the foot contact length ratio, it seems quite important to have gaits that are feasible.
What happens if a gait is not feasible? (e.g. all zeros)

The "Random Pattern Generator" is crucial for training, but the clarity of this section (currently in the appendix) should be improved.

The proposed interface is versatile, but also limited. For example, there is no control over the foot clearance (or ground penetration), CPGs could be an alternative in this case ([3] and [4], CPGs seem to be missing from the related work as well).

A final point is the LLMs used.
As the output of closed LLMs can vary over time [1], it would be good for reproducibility purposes to try with an open source LLM (e.g. [2]) and at least mention the exact version used.


[1] https://arxiv.org/abs/2307.09009
[2] https://chat.lmsys.org/
[3] https://arxiv.org/abs/2209.07171
[4] https://arxiv.org/abs/2211.00458


### Questions/Remarks
- Fig1: what is the intermediate pattern? (between trotting and bounding, how was it generated?)
- "Improves the naturalness of the robot’s movement and helped shrink the sim-to-real gap" -> please show failure
- Only the desired foot pattern is given as input? no information about the current position in the cycle? in that case this would probably break the Markov assumption for the contact reward and make it harder to realize the desired gait.
- The paper claims to allow low-level commands, but in practice it is not that low-level, the LLM generates the foot contact pattern, the rest is done by RL + PD for the low-level part.
- Are unstructured/vague instructions really wanted? Especially since the evaluation is subjective, it seems that this part could be left in the appendix.

**Quality Of The Limitations Section:**

Additional details required

**Questions For Rebuttal:**

See review.

**Robotics Focus:**

Sufficient demonstration on hardware

**Summary Of Paper:**

The paper proposes a method to control quadruped gaits using natural language.
It uses foot contact patterns as an interface and trains a RL controller in simulation to realize the gaits.
The resulting approach is evaluated on the real robot by comparing the generated gaits with the expected output.

**Summary Of Recommendation:**

The idea is interesting, but I would currently recommend rejecting the paper, several important clarifications are needed.

---

> ### Author Response · Authors · 2023-08-15
> **About our responses**
>
> Dear Reviewer,
>
> The author-reviewer discussion phase will end soon. We haven't received your further comments or questions, may we assume that we have addressed all your concerns? In that case, will you consider adjusting the recommendation score? Thank you!

---

### Official Review · Reviewer_NnfC · 2023-07-22

**Confidence:** 4
**Originality:** Good
**Technical Quality:** Very Good
**Clarity Of Presentation:** Very Good
**Impact:** 3

**Recommendation:**

Weak Accept: I recommend accepting the paper, but will not argue for my recommendation if the majority of other reviewers have a different opinion.

**Review:**

The paper is well organized and explained, relatively straightforward to follow.

I’m confused about the “random pattern generator”. I’m understanding it can output a desired contact pattern (sequence of contacts for all four legs) from a set of predefined possible templates ({BOUND, TROT, PACE, STAND_STILL, STAND_3LEGS}). Is this correct? I’m not sure why it is called “random”?

Please clarify the naming of the random generator, perhaps there is a detail I’m not seeing.

How many example blocks (Fig. 3 right) are there in your prompt?

The low level controller for walking was trained using DRL. I was curious if it was necessary to add any specific adaptation to train the policy in the specific set up (i.e. contact sequence pattern and linear velocity inputs) versus the standard training of DRL locomotion policies? I just mean if there was any particular difficulty to make RL converge in your specific set up vs others commonly used ones.

**Quality Of The Limitations Section:**

Limitations are addressed clearly

**Questions For Rebuttal:**

Please clarify the naming of the random generator, perhaps there is a detail I’m not seeing.

How many example blocks (Fig. 3 right) are there in your prompt?

The low level controller for walking was trained using DRL. I was curious if it was necessary to add any specific adaptation to train the policy in the specific set up (i.e. contact sequence pattern and linear velocity inputs) versus the standard training of DRL locomotion policies? I just mean if there was any particular difficulty to make RL converge in your specific set up vs others commonly used ones.


**Robotics Focus:**

Sufficient demonstration on hardware

**Summary Of Paper:**

This paper presents a method to use Large Language Models (LLM) as an intermediary layer between human commands in natural language and the generation of walking patterns for a quadrupedal robot. Through the design of a prompting template, the LLM can convert a language command (e.g. Act as if you have a limping rear left leg) into a sequence of contact instructions and a desired linear velocity  for a low level controller to execute. Using a LLM in this way enables the input instruction to be free form and include aspects such as mood or appearance of the gait.

I find interesting the idea of using a LLM to output this contact sequence representation, which in a way is simple for LLM generation, but that can generate behaviors by using the sequence as a walking pattern.


**Summary Of Recommendation:**

I think the paper has an interesting idea from the conceptual point of view. i.e. using LLM to generate contact sequences for gait generation. The idea can be reused for other problems as an intermediate layer between human user instructions in natural language and behavior generation. Simultaneously, the paper is relatively simple.

---

### Author Response · Authors · 2023-08-08
**Response to All Reviewers**

We thank our reviewers for the insightful comments and questions. We are delighted to learn that reviewers think our paper to be “thought-provoking”, that it “can be reused for other problems as an intermediate layer between human user instructions in natural language and behavior generation”, and that it “looks like an interesting building block towards useful applications”.

We have replied to each reviewer separately, and the following summarizes our responses:

For Reviewer NnfC ([link](https://openreview.net/forum?id=7TYeO2XVqI&noteId=GuhrhxZwuR)), we
1. Answered all the reviewer's questions.
2. Added text in Sec 3.2.2 to highlight what parts in the pattern are randomly sampled.
3. Modified Figure 3 caption to emphasize that it is the exact prompt we use for our method in the experiments, and we use only 3 examples in the prompt (i.e., “Trot slowly”, “Bound in place” and “Pace backward fast”).
4. Added a sentence in sec 3.2.4 to highlight the importance of rebalancing the ratios of the 5 gait types, and deferred the details in Appendix C.2.

For Reviewer 8oYu ([link](https://openreview.net/forum?id=7TYeO2XVqI&noteId=e7kQXX5MAy)), we
1. Answered all the reviewer's questions.
2. Addressed concerns in the reviewer's comments.
3. Added text in Figure 1 caption to explain the desired foot contact pattern in between the commands “Trot forward slowly” and “Good news, we are going to a picnic this weekend!”.
4. Added text in Sec 4.1 and Appendix D to elaborate on the objective evaluation process, we also modified the caption for Table 1.
5. Rewrote Appendix A to clarify how our random pattern generator works.
4. Added possible ways for more versatile representation in Sec 5, and cited related works.

For Reviewer Ztd6 ([link](https://openreview.net/forum?id=7TYeO2XVqI&noteId=0fXJUIPTQg)), we
1. Answered all the reviewer's questions.
2. Expressed our thoughts on the paper's utility.
3. Conducted extra experiments to show the failure cases and attached videos.

For Reviewer erUF ([link](https://openreview.net/forum?id=7TYeO2XVqI&noteId=p4sv2vZp8H)), we
1. Answered all the reviewer's questions.

We hope these address our reviewers’ concerns, and should the reviewers have further comments/questions, we are happy to discuss them.

---

### Decision · Program_Chairs · 2023-08-30

**Decision:**

Accept (Poster)

**Comment:**

## Summary of Paper
The paper proposes a new method to connect language commands via LLMs to quadruped locomotion by utilizing foot placement patterns.

## Summary of Reviews
The paper is well written and has a neat idea. There were a few unclear points and the reviewers had some concerns about the generality of the method (both for other quadruped tasks and more general tasks).

## Influence of Rebuttal
The reviewers appreciated the modifications and clarifications as well as the added experiments on the failure cases. Reviewer 8oYu also reacted during the AC&reviewer discussion phase (not visible to authors, hence copied below). The major concerns of the reviewers have all been addressed.

## Suggestions for Improvement
Add the clarifications to the final version of the paper (largely done) and please also include the new failure cases in the paper. Making the results (more) reproducible is highly encouraged.

## Reviewer 8oYu's reply

### Thanks for the replies

Hello,

thank you for the replies and the edits.

> We submitted our prompt and command to llama-2-13b-chat and alpaca-13b. However, they both failed to give answers in the correct format. In terms of reproducibility, although, unfortunately, we didn’t use a fixed version of GPT-4 for our experiments, the current version still reproduces the results in the paper.

I raised my score to "weak accept" but I will let the AC decide, as a critical aspect for me is reproducibility, and the current reproducibility of the paper is borderline: open sourcing the code for training/evaluation (adding it to the supplementary material) and managing to reproduce the results with an open model would be a great addition.